Medical Imaging with Deep Learning 2023

# High Frequency Structural MRI Signal conditioned MRA Synthesis with Denoising Diffusion probabilistic Model

Haoyu Lan                                          HAOYULAN@USC.EDU
Kirsten M. Lynch                          KIRSTEN.LYNCH@LONI.USC.EDU
Arthur W. Toga                                         TOGA@USC.EDU
Jeiran Choupan                                     CHOUPAN@USC.EDU
*Laboratory of Neuro Imaging, USC Mark and Mary Stevens Neuroimaging and Informatics Institute,*
*USC Keck School of Medicine, University of Southern California, Los Angeles, California, USA*

## Abstract

Magnetic resonance angiography (MRA) allows for the non-invasive visualization of vasculature in the human body and has been widely used in the hospitals to identify aneurysms and the location of a stroke. Generating MRA using the commonly available T1-weighted (T1w) MRI modality would broaden the possibilities for studying vasculature because T1w is commonly acquired in most neuroimaging datasets, while MRA is not. In this work, we propose a method using the statistical generative model called denoising diffusion probabilistic model (DDPM) to tackle the MRA synthesis task. Our experiment shows that by diffusing the high frequency signal, which explains the major signal difference between MRA and T1w, DDPM could successfully synthesize MRA with good quality. The proposed method also conditioned score-matching estimation with the high frequency signal of the T1w modality, which enables the accurate one-to-one synthesis between MRA and T1w.

**Keywords:** MRA, diffusion probabilistic model, high frequency signal, synthesis, vasculature

## 1. Introduction

Magnetic resonance angiography (MRA) is one of the commonly used neuroimaging modalities to visualize cerebrovascular anatomy and vessel abnormalities. As a non-invasive technique, MRA is a safe alternative to traditional angiography methods and can aid in the diagnosis and treatment planning for vascular conditions, such as stroke and aneurysms. Additionally, recent evidence has implicated the cerebrovascular system in many other neurological conditions, such as Alzheimer's disease, and is a crucial component of the waste clearance system. Therefore, improved visualization of brain vasculature can provide critical insight into overall brain health. However, compared to the common T1-weighted (T1w) magnetic resonance imaging (MRI) modality, MRA is generally not acquired in large public datasets, which hinders the study of vasculature in large populations. Statistical generative models like generative adversarial networks (Olut et al., 2018) has shown the power to handle the neuroimaging modality synthesis task. In this work, we propose to use the recent state-of-the-art generative model denoising diffusion probabilistic model (DDPM) (Ho et al., 2020) to synthesize MRA using T1-weighted modality by only diffusing the high frequency signal in the imaging space.

## 2. Methods

We used MRA and T1w images acquired from 18 healthy subjects in the TubeTK dataset (Aylward and Bullitt, 2002) (https://public.kitware.com/Wiki/TubeTK/Data) for the model training and testing (acquisition parameters – MRA: 128 axial slices, 448 x 448 matrix size, 0.5x0.5x0.8 mm3 voxel size; T1w: 170 axial slices, 256 x 256 matrix size, 1x1x1 mm3 voxel size). Each T1w image was spatially aligned and resampled to the corresponding MRA image in native space using FreeSurfer (Fischl, 2012), so that two images are properly registered. Both MRA and T1w modalities were normalized using min-max normalization with maximum intensity thresholds 1000 for MRA and 700 for T1w. Of the 2124 total paired axial slices available from the MRA and T1w images of all subjects, 1700 randomly selected slices were used for training and validation and 424 slices were used for testing.

For the same subject, T1w MRI and MRA share similar low frequency signals representative of gross anatomical features, such as brain shape and volume (Figure 1 A.). Most of the signal differences between T1w and MRA could be explained by the high frequency signal (Figure 1 A.). Inspired by DDPM (Ho et al., 2020) and latent conditional diffusion model (Rombach et al., 2022), we propose to tackle the MRA synthesis problem by diffusing the high frequency signal through the high frequency T1w signal conditioned diffusion process (Figure 1 B.). We designed the diffusion process with 100 steps and the variance of high frequency gaussian noise ranges between 0.001 and 0.2. Score matching model has been implemented following the original DDPM (Ho et al., 2020).

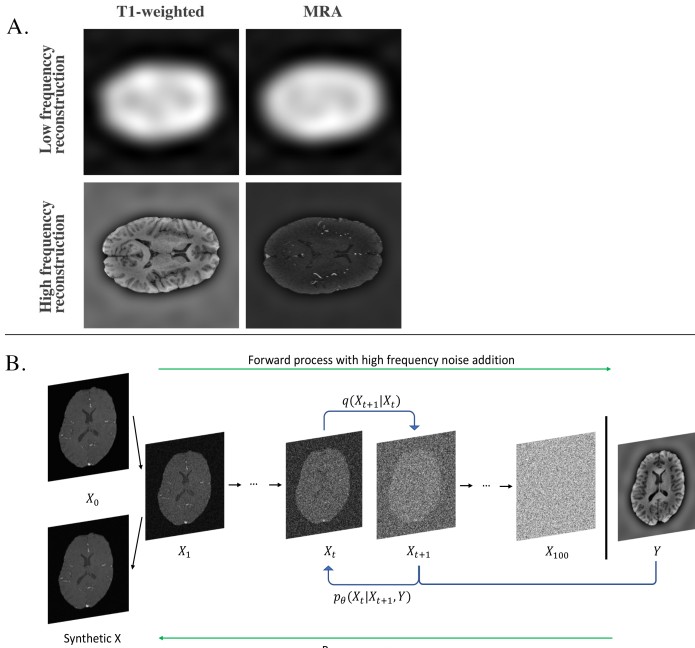

Figure 1: Low frequency and high frequency signal reconstructed modalities and proposed diffusion process.

## 3. Results and discussion

Qualitative assessment of synthetic MRA is shown in Figure 2. By limiting the diffusion process on the high frequency signal space and conditioning the score-matching estimation on the high frequency signal of the T1w, the proposed method synthesizes MRA from the corresponding T1w modality with only a few denoising diffusion steps and high synthesis quality.

In this work, we aim to synthesize MRA modality using high frequency structural T1w MRI as the condition. Most recent successful generative model DDPM has the advantage of stable training compared to GAN model and guaranteed convergence with small variance noise for each diffusion step. As results shown in Figure 2, by focusing the diffusion process on the high frequency signal, the diffusion model could generate MRA with accurate vessel distribution and morphology for the given T1w modality. Thorough model evaluation and comparison will be our next work and we aim to unlock the potential of generative model applications in large-scale neuroimaging datasets and to have neuroimaging research on neurological disease benefit from this application.

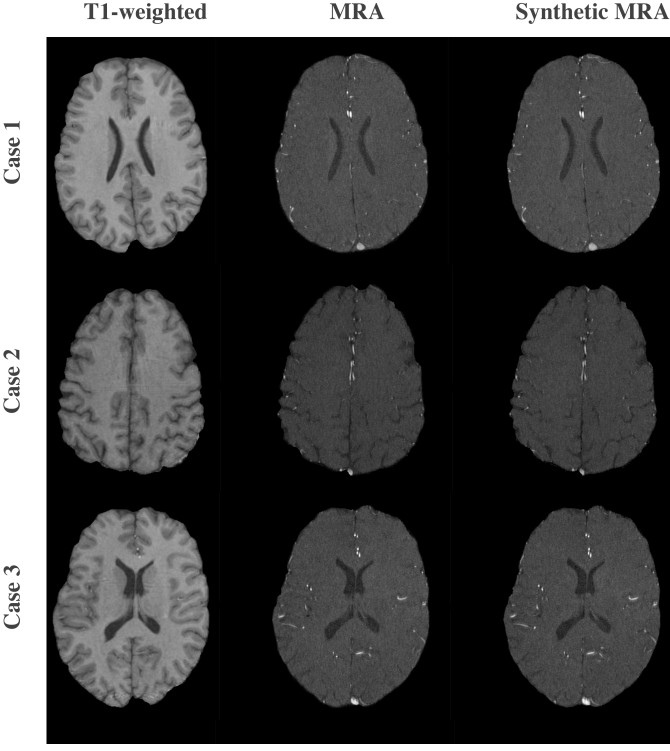

Figure 2: Qualitative assessment of MRA synthesis in axial view.

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
