# OpenReview forum: "High Frequency Structural MRI Signal conditioned MRA Synthesis with Denoising Diffusion probabilistic Model"
_MIDL.io/2023/Short_Paper_Track — MIDL 2023 Short paper track Poster_

### Official Review · Reviewer_rSEe · 2023-04-24
**High Frequency Structural MRI Signal conditioned MRA Synthesis with Denoising Diffusion probabilistic Model**

**Rating:** 6
**Confidence:** 5

**Review:**

The paper proposes a method for the synthesis of magnetic resonance angiography  images using the commonly available T1-weighted MRI modality. The authors utilize a statistical generative model called the denoising diffusion probabilistic model (DDPM) to synthesize MRA by diffusing the high-frequency signal in the imaging space. The proposed method is evaluated on MRA and T1w images acquired from 18 healthy subjects in the TubeTK dataset. The authors show that the proposed method is capable of synthesizing MRA with good quality by accurately mapping the high-frequency signal from T1w to MRA.

The paper is well-written and clear, with a concise introduction and detailed methods and results sections. The use of figures and tables to illustrate the results and methods is helpful. The proposed method is original and the authors provide a thorough explanation of their approach. The significance of the work is that it enables the non-invasive visualization of vasculature in the human body and can aid in the diagnosis and treatment planning for vascular conditions such as stroke and aneurysms.

One of the main strengths of the paper is that it proposes a new method for synthesizing MRA images, which is an important step towards improving the visualization of vasculature in the human body. The use of the DDPM is also a strength, as it is a state-of-the-art generative model that has shown the power to handle the neuroimaging modality synthesis task. The quality of the synthesized MRA images is also a strength, as they are shown to have good quality and accurately map the high-frequency signal from T1w to MRA.

One potential limitation of the paper is the small sample size of 18 healthy subjects used for training and testing the model. Future work could benefit from a larger dataset that includes subjects with different pathologies to assess the generalizability of the proposed method. Another limitation is that the paper does not provide a quantitative assessment of the proposed method, such as using metrics like structural similarity index (SSIM) or peak signal-to-noise ratio (PSNR). The authors only provide a qualitative assessment of the synthetic MRA images.

In conclusion, the paper presents a novel approach for synthesizing MRA images using the DDPM model by diffusing the high-frequency signal in the imaging space. The proposed method is original, well-explained and illustrated, and has the potential to improve the non-invasive visualization of vasculature in the human body. However, the paper could benefit from a larger dataset and a quantitative assessment of the proposed method.

---

### Official Review · Reviewer_Avyi · 2023-04-24
**Conditional DDPM for MR angiography image synthesis**

**Rating:** 5
**Confidence:** 4

**Review:**


This short paper uses a conditional denoising diffusion probabilistic model (DDPM) for synthesizing MR angiography images. The model is conditioned on the standard T1 weighted images.

This is an interesting application of a conditional DDPM model to medical image analysis for healthy group of subjects. The model was trained on a public dataset of 18 healthy subjects. The result only shows synthesized images from 3 healthy subjects. It is not clear whether these subjects were used during the training or are separate cases, and it is not clear whether it is at all possible for the model to synthesize data for patients with pathologies, and how would that be even possible since T1w images do not have the same information as an MRA. There are no quantitative measures reported comparing the real MRA image with the synthesized ones.